# Exogenous melatonin promotes seed germination and osmotic regulation under salt stress in cotton (*Gossypium hirsutum* L.)

**Li Chen**[1,2], **Liantao Liu**[2,3], **Bin Lu**[4], **Tongtong Ma**[1,2], **Dan Jiang**[1,2], **Jin Li**[1,2], **Ke Zhang**[2,3], **Hongchun Sun**[2,3], **Yongjiang Zhang**[2,3], **Zhiying Bai**[1,2]*, **Cundong Li**[2,3]*

**1** College of Life Science, Hebei Agricultural University, Baoding, China, **2** Key Laboratory of Crop Growth Regulation of Hebei Province, Hebei Agricultural University, Baoding, Hebei, China, **3** College of Agronomy, Hebei Agricultural University, Baoding, China, **4** College of Landscape and Tourism, Hebei Agricultural University, Baoding, China

* zhiyingbai@126.com (ZB); auhlcd@163.com (CL)

**Data Availability Statement:** All relevant data are within the manuscript and its Supporting Information files.

## Abstract

Melatonin (MT; *N*-acetyl-5-methoxytryptamine) is an amine hormone involved in abiotic stress resistance. Previous studies have confirmed that melatonin can promote seed germination, mediate physiological regulation mechanisms, and stimulate crop growth under stress. However, the osmotic regulation mechanism by which exogenous melatonin mediates salt tolerance in cotton is still largely unknown. To investigate the effect of salt stress on melatonin concentration in germinating cotton seeds, we analyzed melatonin content over time during seed germination under different treatments. Melatonin content reached its minimum at day 6, while cotton germination rates peaked at day 6, indicating melatonin content and seed germination are correlated. Then we investigated the effects of 10–100 μM melatonin treatments on membrane lipid peroxides and osmotic adjustment substances during cotton seed germination under salt stress. Salt stress led to electrolyte leakage (EL) as well as accumulations of hydrogen peroxide ($H_2O_2$), malondialdehyde (MDA), organic osmotic substances (i.e., proline, soluble sugars), and inorganic osmotic substances (i.e., $Na^+$, $Cl^-$). Meanwhile, the contents of melatonin, soluble proteins, and $K^+$ as well as the $K^+/Na^+$ balance decreased, indicating that salt stress inhibited melatonin synthesis and damaged cellular membranes, seriously affecting seed germination. However, melatonin pretreatment at different concentrations alleviated the adverse effects of salt stress on cotton seeds and reduced EL as well as the contents of $H_2O_2$, MDA, $Na^+$, and $Cl^-$. The exogenous application of melatonin also promoted melatonin, soluble sugar, soluble proteins, proline, and $K^+/Na^+$ contents under salt stress. These results demonstrate that supplemental melatonin can effectively ameliorate the repression of cotton seed germination by enhancing osmotic regulating substances and adjusting ion homeostasis under salt stress. Thus, melatonin may potentially be used to protect cotton seeds from salt stress, with the 20 μM melatonin treatment most effectively promoting cotton seed germination and improving salt stress tolerance.

**Funding:** This work was supported by the National Science Foundation of China (No. 31871569 and 31571610).

**Competing interests:** The authors have declared that no competing interests exist.

**Abbreviations:** MT, melatonin; GR, germination rate; $H_2O_2$, hydrogen peroxide; MDA, malondialdehyde; EL, electrolyte leakage; ROS, reactive oxygen species; IAA, indole-3-acetic acid.

## Introduction

Salt stress is a critical environmental factor that limits the agricultural productivity, survival, and geographical distributions of plants [1]. Approximately 950 million $hm^2$ of land in the world is affected by salinization, of which about 90 million $hm^2$ of impacted areas occur in China alone, placing great importance on the study of salt stress effects on plant growth and development [2]. Under salt stress, the oxidative reaction of free radicals in membrane lipids leads to the accumulation of more reactive oxygen radicals and hydrogen peroxide in plants, which causes cross-linking polymerization of proteins, nucleic acids, and other biomolecules as well as damage to the membrane system, which in turn increases malondialdehyde (MDA) content and electrolyte leakage (EL) as well as lipid peroxidation. The plant membrane system is one of the first sites initially injured by stress conditions, including high salinity [3–5]. Plants mainly resist salt stress damage through a series of physiological activities that includes osmotic regulation, ion transport, and hormone content change [6]. Osmotic regulators in plants mainly include organic osmotic regulators (proline, soluble sugar, soluble protein) and inorganic osmotic regulators ($Na^+$, $K^+$, $Cl^-$), which can together increase cell fluid concentration and reduce osmotic potential as well as maintain intracellular homeostasis and enhance plant resistance to salt stress [7].

Melatonin is a well-known amine hormone, synthesized in chloroplasts and mitochondria, and it is found in most plants and animals [8–10]. Thus, endogenous melatonin is expected to occur in cotton seeds. Moreover, melatonin is an extremely efficient antioxidant that can effectively suppress $H_2O_2$ production through enhancing activities of CAT, POD, and APX under drought stress, improving antioxidant defense systems through MT-induced generation of NO and by lowering MDA and $H_2O_2$ levels, and regulating various physiological process in plants [11–13]. Accordingly, melatonin enhances plant growth and development under abiotic stresses (such as high salt, heavy metals, etc.) and acts as an osmotic regulatory substance in plants, enabling the maintenance of ion homeostasis and growth regulation [14]. In recent years, more plant science research has been focused on the role of melatonin, which is closely related to indole-3-acetic acid (IAA) in its chemical structure and metabolic pathways [15]. Previous studies have shown that exogenous melatonin can maintain high photosynthetic efficiency in tea plants through its effects on antioxidant systems against abiotic stress, which that enhance salt and cold tolerance [16]. The content of hydrogen peroxide and malondialdehyde in cucumber seedlings obviously increased under irrigation with salt solution, which indicated that salt treatment damaged cell membranes [17]. Furthermore, exogenous melatonin application can significantly reduce the accumulation of hydrogen peroxide and increase the activities of antioxidant enzymes [18]. Melatonin treatment has also been shown to significantly reduce electrolyte leakage in tomato plants under cadmium stress; however, it had no significant effect on tomatoes under normal conditions [19]. Salt stress severely inhibits the growth of soybean seedlings, and the application of exogenous melatonin can increase the soluble protein content of seedlings [20]. The appropriate application of melatonin has been shown to promote not only soluble protein content in *Malus hupehensis* but also the accumulation of soluble sugar in kiwifruit leaves, while increasing cell fluid concentrations and resistance to stress [21, 22]. Under salt stress, accumulation of $Na^+$ under salt stress of plants leads to ion imbalances and toxicity, with $K^+$ content decreasing as exogenous melatonin promotes the absorption of $K^+$, suggesting that melatonin can regulate ion homeostasis or gene expression responses under salt stress to mitigate damage caused by stress [23–26].

Cotton (*Gossypium hirsutum* L.) is a major global crop, and it is widely cultivated in China. Under increasing soil salinization, the growth, yield, and quality of cotton harvests have been seriously affected. Seed germination is perhaps the most important and most complex process

in the plant growth cycle, as it directly affects the development of cotton plants, which ultimately affects yields. Seed germination can be divided into three periods. First, seeds absorb water from the environment. In the second stage, seed germination is stimulated by various enzymes and hormones that prepare seeds for germination. Finally, the radicle breaks through the seed coat and lengthens. Accordingly, seed germination involves many processes [27]. While the response mechanism of cotton seed germination and seedlings to salt stress have been investigated, the mechanism by which exogenous melatonin influences osmotic control and thus the germination process of cotton seeds under salt stress remains unclear. Therefore, different melatonin concentration treatments were employed in this experiment to investigate the effects of exogenous melatonin on physiological activities such as membrane lipid peroxide, osmotic regulators, and ion homeostasis during the germination of cotton seeds under salt stress. This study provides some novel insights into salt tolerance mechanisms modulated by exogenous melatonin in seed germination.

## Materials and methods

### Reagents

All chemicals used in this study were of analytical grade. Melatonin (*N*-acetyl-5-methoxytryptamine) was purchased from Sigma-Aldrich (St. Louis, MO, USA). All other reagents were purchased from Sinopharm Chemical Reagent Beijing Co., Ltd, (Beijing, China).

### Plant material

Cotton (*Gossypium hirsutum* L.) cultivar 'GXM9' seeds (provided by Guoxin Rural Technical Service Association of Hejian City, China) were used in the study. The experiment was conducted in the greenhouse facilities of Hebei Agricultural University in Baoding (38.85˚N, 115.30˚E) City, Hebei Province from September 2018 to May 2019.

### Seed germination

Cotton seeds with full seed coats and of consistent size were disinfected with 75% ethanol for 17 min, rinsed in distilled water four times, and dried in a cool and ventilated area. All seeds were randomly divided into six groups, and the experimental treatments were as follows: (1) Control (Con), primed with water without salt treatment; (2) NaCl, primed with water and then treated with salt (150 mM NaCl, screened by the pretest); (3) $MT_{10}$+NaCl, primed with 10 μM MT (melatonin) solution and then treated with salt (150 mM NaCl); (4) $MT_{20}$+NaCl, primed with 20 μM MT solution and then treated with salt (150 mM NaCl); (5) $MT_{50}$+NaCl, primed with 50 μM MT solution and then treated with salt (150 mM NaCl); and (6) $MT_{100}$+NaCl, primed with 100 μM MT solution and then treated with salt (150 mM NaCl). Three replicates of 500 seeds were used for each treatment.

The cotton seeds were soaked in distilled water or one of the different concentrations of melatonin solutions for 24 h in 15-cm-diameter Petri dishes containing filter paper (Whatman International Ltd, Maidstone, UK), dried, and restored to their initial water content over the course of about 2 d. Seeds were then placed in a light culture box (25˚C) and cultured in dark conditions for 6 d. Seeds were examined every two days and were considered germinated when the seed coat was broken and a radicle was visible. Germinated seeds were sampled from each treatment, rapidly frozen in liquid nitrogen, and stored at -80˚C until analysis. All experiments were conducted in triplicate.

## Determination of cotton seed germination rate

The number of cotton seeds germinated was recorded at 2, 4, and 6 d after seeds were placed in incubators. The following equation was used to calculated the germination rate:

Germination rate = total germinated seeds / total seeds × 100%.

## Determination of melatonin content

Melatonin was extracted from cotton seeds using the Plant MT ELISA KIT (Shanghai MLBIO Biotechnology Co. Ltd., Shanghai, China) according to the manufacturer's instructions. The samples to be tested were incubated with antibodies and measured at 450 nm with a microplate reader (Bio Tek Instrument, Inc., Winooski, VT, USA).

## Determination of $H_2O_2$, MDA, and EL

Hydrogen peroxide content was determined according to the method used by Sun et al. [28] with some slight modifications. Two grams of cotton seeds were ground in a mortar, with 2 ml of acetone, followed by centrifugation at 10,000 rpm for 10 min. To the supernatant (i.e., hydrogen peroxide extract), acetone was added to reach a total volume of 3 ml. Then, 1 ml of extract, 3 ml of extraction agent, and 5 ml of distilled water were mixed and then centrifuged at 5000 rpm for 1 min, after which, 2 ml of working reagent was mixed. The light absorption value was then measured at 560 nm using a spectrophotometer (UV2450, Shimadzu Corp., Kyoto, Japan).

MDA content was measured according to the method described by Cui et al. [29] with some slight modifications. Cotton seeds were fully ground in pH 7.8 phosphate buffer, followed by centrifugation at 6000 rpm for 10 min. Two milliliters of supernatant was added to the scale test tube, to which 1 ml of 0.5% thiobarbituric acid and 3 ml of 5% trichloroacetic acid solution were added. The solution was heated in a boiling water bath for 10 min and then cooled rapidly. After centrifugation at 6000 rpm for 10 min, the light absorption value was measured at 532 nm and 600 nm using a spectrophotometer (UV2450, Shimadzu Corp.) with distilled water as the blank and 100% light transmittance.

EL was measured according to the assay described by Wu et al. [30] with some slight modifications. First, 0.1 g of fresh sample material was placed in a glass container, to which 30 ml of deionized water was added. The container was placed into a vacuum dryer, and air was extracted from the cells for 1 h. After standing for 5 min, sample conductivity was measured. Samples were sealed with foil, boiled in water for 30 min, and cooled, after which conductivity was measured. Finally, the relative conductivity was calculated.

## Determination of osmotic regulators proline, soluble sugar, and soluble protein

The determination of proline content was measured according to the method by Bates et al. [31] with some slight modification. First, 0.3-g samples were cut into pieces and added to a mortar, to which an appropriate amount of 80% ethanol and a small amount of quartz were added prior to grinding the tissue into a homogenate. Then, the volume was filled with 80% ethanol to 25 ml and incubated in an 80°C watered bath for 20 min, after which 0.4 g of artificial zeolite and 0.2 g of activated carbon were added. Samples were subsequently oscillated and filtered, and 2 ml of the above extraction solution was transferred into a test tube, to which 2 ml of glacial acetic acid and 2 ml of indanone were added before being heated in boiling water for 15 min. After cooling, the light absorption values of the samples were measured at 520 nm using a spectrophotometer (UV2450, Shimadzu Corp.).

Soluble sugar content was determined using the anthraquinone colorimetric method [32]. Then, 0.3-g samples and 9 ml of distilled water were added to test tubes that were placed in a boiling water bath for 20 min and cooled. One milliliter of the supernatant and 5 ml of sulfuric acid-anthrone reagent were mixed, placed in a boiling water bath for 10 min, and cooled. The light absorption value of each sample was measured at 620 nm using a spectrophotometer (UV2450, Shimadzu Corp.).

Soluble protein content was determined using Coomassie brilliant blue. First, 0.3-g samples were ground into a homogenate with 5 ml of pH 7.8 phosphate buffer. The supernatant was centrifuged at 4000 rpm for 10 min, and to 0.1 ml of the supernatant, 9 ml of distilled water and 5 ml of Coomassie brilliant were added, followed by centrifugation again and then sample oscillation. After standing for 5 min, the light absorption value was measured at 620 nm using a spectrophotometer (UV2450, Shimadzu Corp.) and distilled water with Coomassie brilliant blue was used as the blank control.

## Determination of ion content

To determine the $Na^+$ and $K^+$ contents of cotton seeds, well-ground samples were heated to 500°C for 6 h in a muffle furnace. To the resulting white ash, 5 ml of 2 M hot HCl was added, and the final volume was raised to 50 ml by adding distilled deionized $H_2O$. The above solution was measured for atomic absorption (ZA3000, Hitachi, Ltd., Tokyo, Japan), and the obtained data were subjected to a final calculation. To determine $Cl^-$ content, we used a $Cl^-$ kit obtained from Nanjing Jiancheng Company (Nanjing, China).

## Statistical analysis

The experiment was conducted according to a completely randomized design with three replicates. Analysis of variance (ANOVA) was conducted with SPSS software 21.0 (IBM Corp, Armonk, NY, USA). Differences among treatment means were assessed using Tukey's honest significant different test considered significant at a $p < 0.05$ threshold.

## Results

### Exogenous melatonin promotes cotton seed germination under salt stress

Seed germination is the most important stage in the life course of seeds, and it provides the nutritional basis for the growth and development of seeds into seedlings. We conducted an extensive set of germination assays using cotton seeds to examine the effects of different concentrations of melatonin on seed germination under treatment with 150 mM NaCl (screened by the pretest). As shown in Fig 1, as the germination assay continued, the cotton germination rate continued to increase, reaching its maximum at 6 d. The seed germination rate reached 89.00% under normal conditions (Con), while the germination rate was only about 73.30% under salt stress (NaCl) at 6 d, indicating that NaCl indeed inhibited cotton seed germination. When different concentrations of melatonin were applied, the cotton germination rate exhibited different trends. As melatonin concentrations increased, the cotton germination rate first rose and then decreased, indicating that melatonin affected seed germination in a dosage-dependent manner under salt stress. Low melatonin concentrations (i.e., $MT_{10}$+NaCl and $MT_{20}$+NaCl treatments) effectively promoted seed germination; however, the effect of high concentrations of melatonin (i.e., $MT_{50}$+NaCl and $MT_{100}$+NaCl treatments) on seed germination was not obvious. Among the treatments examined, the 20 μM melatonin pretreatment had the strongest effect in promoting cotton seed germination at different periods.

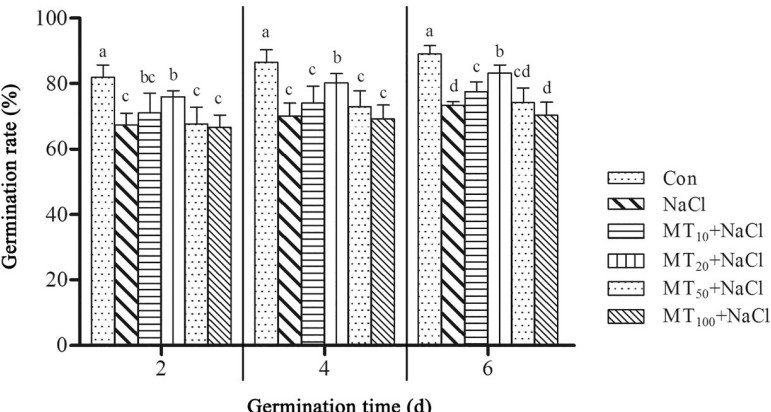

**Fig 1. Effects of exogenous melatonin on GR in cotton seeds under salt stress.** The data are the means of six replicates (±SE), and treatments with different letters are significantly different at a $p < 0.05$ threshold.

## Melatonin content of seed germinating under different treatments

The melatonin contents of cotton seeds under different treatments were determined. As show in Fig 2, as the germination assays continued, melatonin content in all treatments decreased and reached a minimum at day 6, which indicated that melatonin played a regulatory role in seed germination. At days 2, 4, and 6, the melatonin contents of NaCl stress seeds were significantly lower, by 9.28%, 15.48%, and 21.39% respectively, compared with those of Con seeds, which indicated that NaCl inhibited melatonin synthesis. After applying different exogenous concentrations of melatonin, the melatonin content was significantly higher than that of NaCl seeds. Notably, the melatonin content under the $MT_{10}+NaCl$ treatment was lowest; however, the melatonin content under the $MT_{100}+NaCl$ treatment was highest among all treatments, indicating that exogenous melatonin promoted the accumulation of endogenous melatonin.

## Exogenous melatonin mitigates membrane permeability during cotton seed germination under salt stress

**Exogenous melatonin reduces $H_2O_2$ content in cotton seeds under salt stress.** Under stress, more $H_2O_2$ is produced, and these reactive oxygen species can produce toxic effects

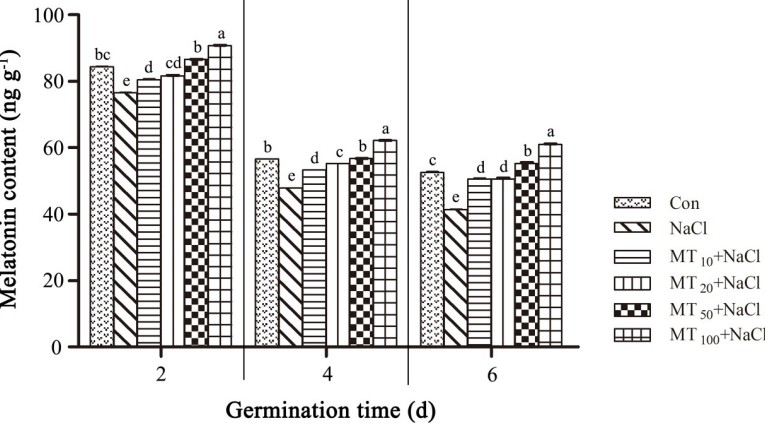

**Fig 2. Melatonin content of seeds germinating under different treatments.** The data are the means of four replicates (±SE), and treatments with different letters are significantly different at a $p < 0.05$ threshold.

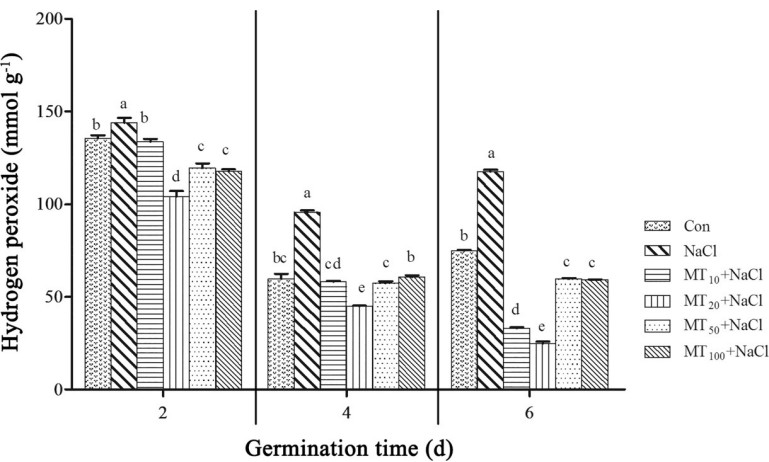

**Fig 3. Effects of exogenous melatonin on $H_2O_2$ content in cotton seeds under salt stress.** The data are the means of three replicates (±SE), and treatments with different letters are significantly different at a $p < 0.05$ threshold.

through cellular aerobic metabolism. As shown in Fig 3, as the germination assays continued, the hydrogen peroxide content of cotton seeds first sharply decreased and then increased. At 2, 4, and 6 d, the $H_2O_2$ contents of NaCl seeds were 6.28%, 60.54%, and 57.26% higher, respectively, compared with those of Con seeds, which indicated that $H_2O_2$ gradually accumulated in cotton seeds under salt stress. After applying different exogenous concentrations of melatonin, the $H_2O_2$ content was significantly lower than that of NaCl seeds. Notably, the $H_2O_2$ content decreased the most under the $MT_{20}+NaCl$ treatment, and it was 27.67%, 52.98%, and 78.69% lower than that of the NaCl treatment at 2, 4, and 6 d, respectively, indicating that the effect of 20 μM melatonin most obviously inhibited $H_2O_2$ accumulation.

**Exogenous melatonin decreases MDA accumulation in cotton seeds under salt stress.** As shown in Table 1, during the seed germination assay, MDA contents under the NaCl treatment were significantly higher than those under the Con treatment, which were 13.36%, 23.70%, and 34.19% higher than those under the Con treatment at 2, 4, and 6 d, respectively, indicating that salt stress led to membrane lipid oxidation and increased MDA content. The MDA content of cotton seeds under exogenous melatonin application was significantly lower than that under the NaCl treatment. Among treatments, the 20 μM melatonin-treated seeds at 2, 4, and 6 d had MDA contents 39.46%, 27.62%, and 34.42% lower, respectively, than those of NaCl seeds. As melatonin concentration increased, MDA content increased slightly under $MT_{50}+NaCl$ and $MT_{100}+NaCl$ treatments, indicating that an appropriate melatonin concentration can effectively reduce membrane oxidation injury and protect cell structure stability.

**Table 1. MDA content (μmol $g^{-1}$) of melatonin-pretreated cotton seeds under salt stress.**

| Treatment | 2 d | 4 d | 6 d |
| --- | --- | --- | --- |
| Control (Con) | 1.861±0.049b | 2.006±0.010b | 2.066±0.059b |
| NaCl | 2.110±0.152a | 2.481±0.088a | 2.772±0.104a |
| $MT_{10}+NaCl$ | 1.666±0.063c | 2.001±0.118b | 1.960±0.053bc |
| $MT_{20}+NaCl$ | 1.277±0.038d | 1.796±0.119c | 1.818±0.030d |
| $MT_{50}+NaCl$ | 1.812±0.014bc | 1.870±0.104bc | 1.867±0.061cd |
| $MT_{100}+NaCl$ | 1.868±0.064b | 1.936±0.057bc | 2.045±0.102b |

Cells followed by different letters within a column are significantly different at a $p < 0.05$ threshold.

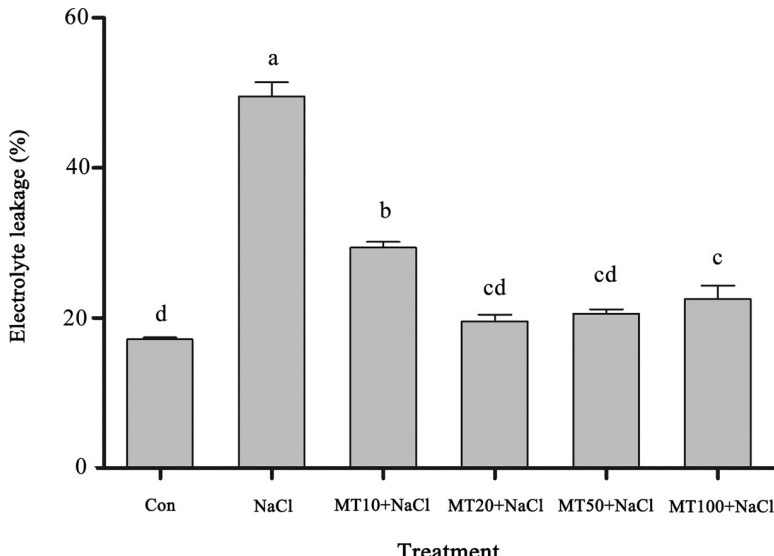

**Fig 4. Effects of exogenous melatonin on EL content in cotton seeds under salt stress.** The data are the means of three replicates (±SE), and treatments with different letters are significantly different at a $p < 0.05$ threshold.

**Exogenous melatonin reduces cotton seed EL under salt stress.** EL is an important index of cell membrane damage. During salt stress, membrane lipid peroxidation damages membranes and increases membrane permeability, resulting in solution extravasation and increased EL. As shown in Fig 4, EL was low under normal conditions (Con), which showed that the cell membranes of cotton seeds were relatively complete. Salt stress significantly increased EL in cotton seeds, which was 188.97% higher than that under the Con treatment. However, the EL of cotton seeds treated with exogenous melatonin was significantly lower. $MT_{10}+NaCl$, $MT_{20}+NaCl$, $MT_{50}+NaCl$, and $MT_{100}+NaCl$ treatments had 40.69%, 60.46%, 58.44%, and 54.44% lower EL, respectively, compared with the NaCl treatment. The EL of cotton seeds under the $MT_{20}+NaCl$, treatment was the lowest of all treatments across different periods, indicating that the appropriate melatonin concentration could effectively alleviate cell membrane damage to cotton seeds caused by salt stress.

## Exogenous melatonin affects organic osmotic substance content during cotton seed germination under salt stress

**Exogenous melatonin affects the proline content of cotton seeds under salt stress.** Proline, an important osmotic regulator in plants, can improve the ability of plants to resist stress. As shown in Fig 5A, proline content decreased overall across the seed germination assay. Under salt stress (NaCl), proline content in cotton seeds decreased obviously, with concentrations of 473.39 µg g$^{-1}$, 154.35 µg g$^{-1}$, and 132.34 µg g$^{-1}$, respectively, 2, 4, and 6 d into the germination assay. The proline contents of NaCl seeds were 21.30% and 3.71% higher and 1.75% lower respectively, compared to Con seeds at 2, 4, and 6 d, respectively, indicating that proline decreased gradually in cells. At 2 d, the proline content was significantly lower under the NaCl treatment than under the $MT_{10}+NaCl$, $MT_{50}+NaCl$, and $MT_{100}+NaCl$ treatments; however, proline contents were significantly higher under the $MT_{20}+NaCl$ treatments, indicating that exogenous melatonin had less effect on proline content initially (i.e., 2 d), while it eventually promoted the accumulation of proline in cotton seeds under salt stress (i.e., 4 and 6 d), which was associated with tolerance to salt stress. Notably, the proline content under the $MT_{20}+NaCl$

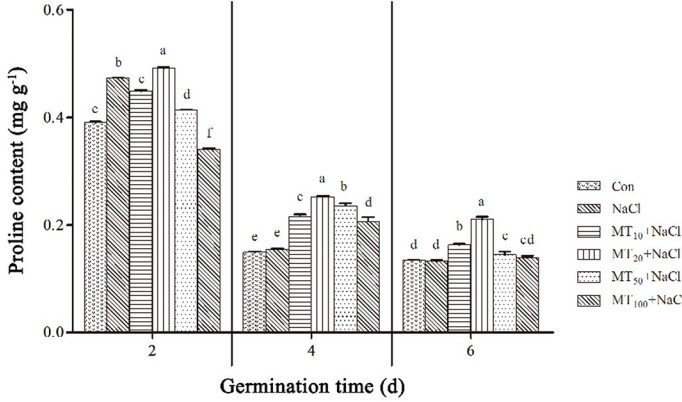

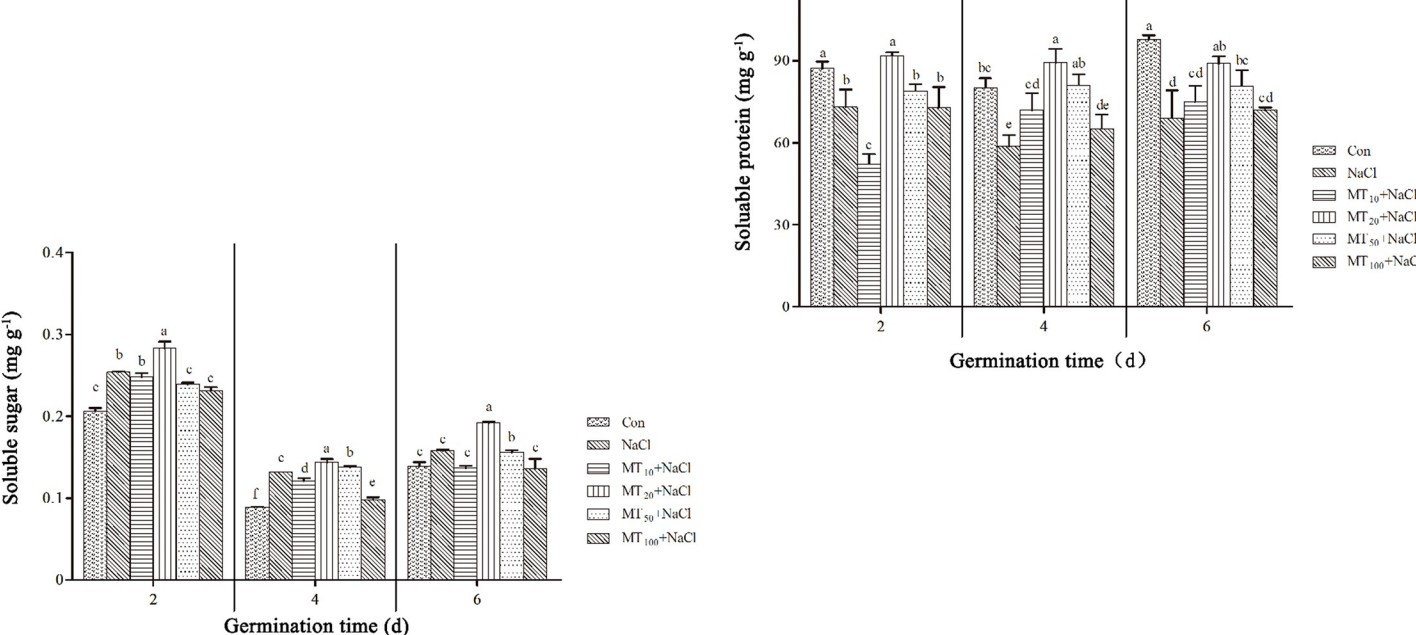

**Fig 5.** Effects of exogenous melatonin on proline (A), soluble sugar (B), and soluble protein (C) contents of cotton seeds under salt stress. The data are the means of three replicates (±SE), and treatments with different letters are significantly different at a $p < 0.05$ threshold.

treatment was the highest across all treatments at different periods, being 3.76%, 63.24%, and 59.25% higher, respectively, at 2, 4, and 6 d compared with the NaCl seeds.

**Exogenous melatonin affects the soluble sugar content of cotton seeds under salt stress.** Soluble sugar is also an important osmotic regulator that provides energy for plant growth and development to ensure a well-functioning metabolism. As shown in Fig 5B, as the assay continues, the soluble sugar content overall decreased and then increased. Compared to Con seeds, the soluble sugar content in NaCl seeds was significantly higher, indicating that plant cells must accumulate some organic matter to cope with salt stress. However, the total accumulated sugar by itself was relatively limited, which may have been insufficient for germination. The soluble sugar content trends of cotton seeds differed among melatonin treatments. $MT_{10}$+NaCl and $MT_{100}$+NaCl seeds had lower sugar content than did NaCl seeds over the

course of the assay, while that of $MT_{20}$+NaCl seeds was significantly higher throughout the assay, by 11.43%, 9.32%, and 21.57%, respectively, at 2, 4, and 6 d. Additionally, the soluble sugar content of cotton seeds under the $MT_{50}$+NaCl treatment was significantly higher than that of NaCl seeds at 4 and 6 d. This indicates that both low and high concentrations of melatonin ($MT_{10}$+NaCl and $MT_{100}$+NaCl, respectively) inhibited the accumulation of soluble sugar in cotton seeds, and intermediate concentrations of melatonin ($MT_{20}$+NaCl and $MT_{50}$+NaCl) were effective in promoting the accumulation of soluble sugar in cotton seeds.

Among treatments, the $MT_{20}$+NaCl treatment had the highest soluble sugar content, indicating it is a suitable concentration of melatonin for improving salt stress resistance in cotton seeds.

**Exogenous melatonin affects soluble protein content of cotton seeds under salt stress.** Soluble proteins are among the main metabolites that accumulate in various species of higher plants in response to salt stress. Fig 5C shows soluble protein content reductions for the NaCl-treated seeds of 16.13%, 26.78%, and 29.49%, respectively, compared with Con seeds at 2, 4, and 6 d. Under all melatonin treatments, except for $MT_{10}$+NaCl, the soluble protein contents were significantly higher than under the NaCl treatment. In particular, the soluble protein content under the $MT_{20}$+NaCl treatment was the highest across all stages. This suggests that melatonin supported macromolecular structure proteins and played a role in maintaining cell stability.

## Exogenous melatonin affects inorganic osmotic regulators in cotton seeds under salt stress

**Exogenous melatonin affects $Na^+$ and $Cl^-$ content of cotton seeds under salt stress.** $Na^+$ toxicity is one of the main components of salt stress in plants. As shown in Fig 6A, at 2, 4, and 6 d, the content of $Na^+$ in NaCl seeds under salt stress was significantly higher, by 21.00%, 32.50%, and 62.11%, respectively, compared with Con seeds, resulting in an ion imbalance. After treatment with various concentrations of melatonin, $Na^+$ content was significantly lower, demonstrating that melatonin slowed the rate of ions entering cells, thereby effectively protecting the cellular structure. $Na^+$ contents under the $MT_{20}$+NaCl treatment were 14.72%, 22.89%, and 28.40% lower at 2, 4, and 6 d, respectively, relative to the NaCl treatment, showing the most obvious $Na^+$ accumulation inhibition among melatonin treatments. Fig 6B shows that not only $Na^+$ content but also $Cl^-$ content increased significantly under salt stress. At 2, 4, and 6 d, the $Cl^-$ content under the NaCl treatment was 152.06%, 54.52%, and 109.90% higher, respectively, compared with Con seeds. The $Cl^-$ content decreased significantly under different concentrations of melatonin. In particular, the $Cl^-$ contents under the $MT_{20}$+NaCl treatment were 46.86%, 31.04%, and 46.14% lower at 2, 4, and 6 d, respectively, relative to the NaCl treatment. The trend in $Cl^-$ content was similar to that in $Na^+$ content, indicating that melatonin may reduce damage caused by $Cl^-$, thus protecting cell membranes.

**Exogenous melatonin affects $K^+$ content and the $K^+/Na^+$ balance of cotton seeds under salt stress.** K is an essential element in plants, and high $K^+/Na^+$ balance can improve salt tolerance. As shown in Fig 6C, the $K^+$ content of cotton seeds decreased significantly throughout the germination assay. Among the different melatonin treatments, $K^+$ content trends over time differed. When treated with 20 μM melatonin ($MT_{20}$+NaCl), $K^+$ content increased by 21.20%, 33.72%, and 4.87% at 2, 4, and 6 d, respectively, compared with NaCl seeds, indicating that an appropriate concentration of melatonin can maintain ion homeostasis and relieve the toxicity of salt stress. As shown in Table 2, salt stress (i.e., NaCl treatment) resulted in significantly lower $K^+/Na^+$ balances, which were 49.19%, 43.30%, and 52.02% lower than those of Con seeds at 2, 4, and 6 d, respectively. However, the $K^+/Na^+$ balance increased significantly

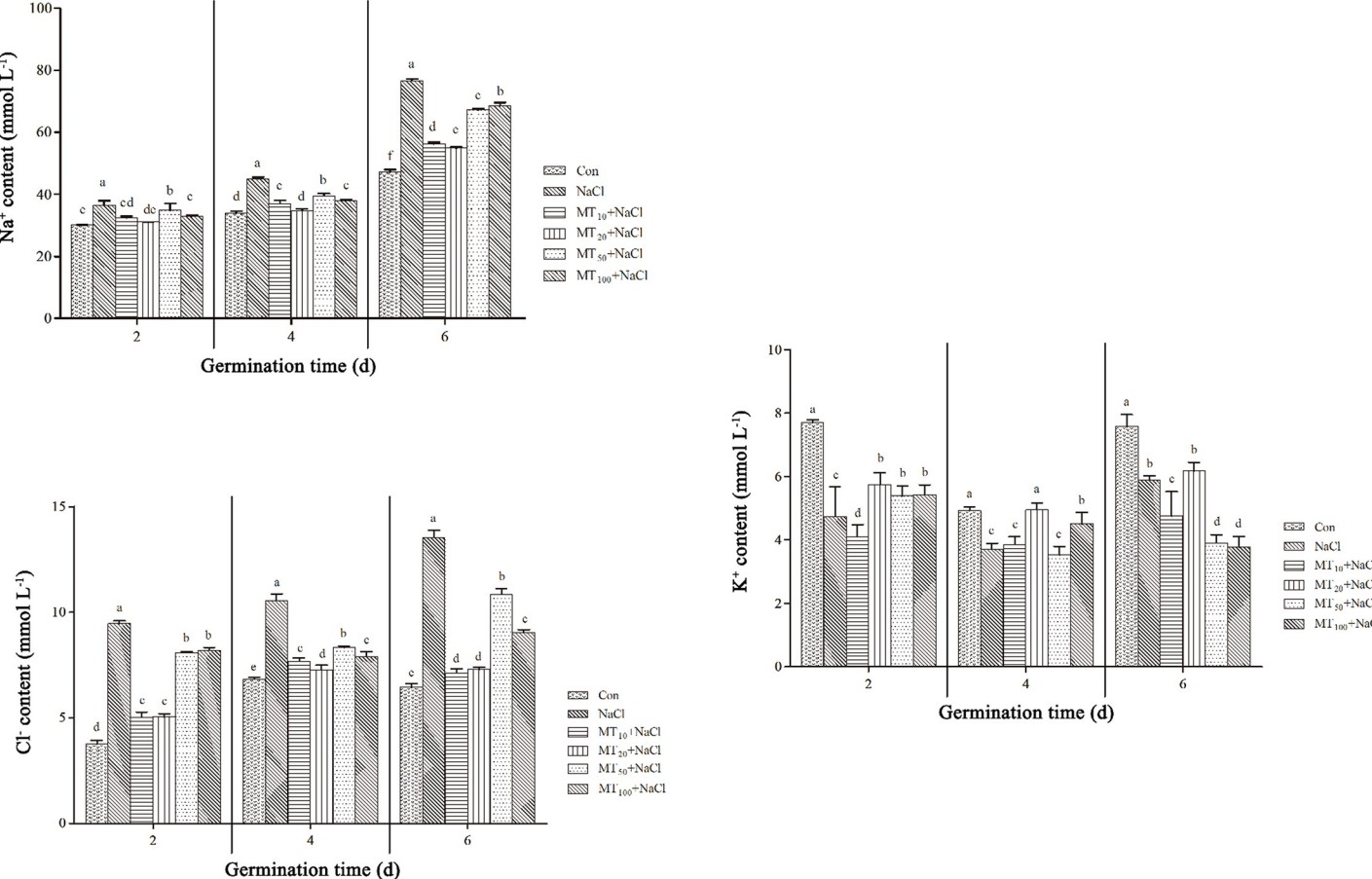

**Fig 6.** Effects of exogenous melatonin on $Na^+$ (A), $Cl^-$ (B), and $K^+$ (C) content in cotton seeds under salt stress. The data are the means of six replicates (±SE), and treatments with different letters are significantly different at a $p < 0.05$ threshold.

after the application of different melatonin concentrations. Notably, $MT_{20}$+NaCl treatment increased the $K^+/Na^+$ balance by 42.21%, 73.44%, and 46.46% compared with NaCl at 2, 4, and 6 d, respectively. Accordingly, appropriate concentrations of melatonin can effectively control the rate of $K^+/Na^+$ intake and maintain a relatively high $K^+/Na^+$ balance, as well as protect intracellular ion homeostasis in cotton seed cells under salt stress.

**Table 2. Effects of melatonin treatment on $K^+/Na^+$ balance in cotton seeds under salt stress.**

| Treatment | 2 d | 4 d | 6 d |
|---|---|---|---|
| Control (Con) | 0.256±0.0036a | 0.145±0.0042a | 0.160±0.0082a |
| NaCl | 0.130±0.0253d | 0.082±0.0047d | 0.077±0.0017c |
| $MT_{10}$+NaCl | 0.127±0.0118d | 0.104±0.0070c | 0.084±0.0138c |
| $MT_{20}$+NaCl | 0.185±0.0126b | 0.143±0.0071a | 0.113±0.0049b |
| $MT_{50}$+NaCl | 0.155±0.0139c | 0.089±0.0077d | 0.058±0.0038d |
| $MT_{100}$+NaCl | 0.165±0.0102c | 0.119±0.0093b | 0.055±0.0051d |

Cells followed by different letters within a column are significantly different at a $p < 0.05$ threshold.

## Discussion

In recent years, abiotic stress has markedly impaired crop yields [33]. Salt is one of the main factors affecting ecological environments and inhibiting crop growth. With the expansion of saline-alkali land areas in China, crop yields and quality have decreased gradually [34]. The salt stress damage to plants is mainly manifested through osmotic stress, ion stress, and antioxidant system and hormone signal transduction, among other phenomena [35, 7]. Various osmotic regulatory substances can be actively accumulated in plants, allowing them to cope with damaged cells and maintain homeostasis between the internal and external environments of cells, thus enabling plants to slowly adapt to salt stress [23]. A similar pattern is observed in salt-stress seeds, but with reduced growth rates and germination rates, indicating salt stress affects germination of cotton seeds. Meanwhile, the contents of $H_2O_2$, EL, and MDA increased and the content of organic osmotic substances in cotton seeds increased, such as proline and soluble sugar, which changed in the seeds upon exposure to saline stress. Moreover, the content of inorganic osmotic regulatory substances increased, including $Na^+$ and $Cl^-$, which changed in the seeds owing to a loss of control under saline stress; however, the content of soluble proteins and $K^+$ as well as the $K^+/Na^+$ balance decreased. Together, these results indicate that salt stress led to membrane lipid peroxidation, which affected cotton seed germination, consistent with the results of Samea-Andabjadid et al. [5] and Castanares et al. [36].

Melatonin is a small molecular hormone found in plants and animals; it is a broad-spectrum growth regulator and antioxidant that resists peroxidation damage to plants caused by stresses such as salt and drought stresses [23]. Abiotic stress can induce changes in the melatonin content of plants [37]. The present study showed that melatonin levels decreased with time in the control (Fig 2), consistent with their consumption by antioxidant action. Salt stress can reduce melatonin levels in cotton seeds (Fig 2); as there are lower reactive oxygen species levels under higher melatonin treatments, it is more likely that it is consumed by antioxidant action of the melatonin, which is consistent with previous findings [17]. However melatonin content significantly increased after the application of exogenous melatonin, indicating that exogenous melatonin could induce endogenous melatonin accumulation [17]. Melatonin content reached its minimum on day 6, while the germination rate peaked at day 6 of the seed germination trial, indicating a relationship between melatonin content and seed germination. During germination, melatonin content decreased while alleviating the inhibitory effects of high salinity. However, melatonin content was not positively correlated with seed germination rate under salt stress. Low melatonin concentrations (i.e., $MT_{10}$+NaCl and $MT_{20}$+NaCl treatments) effectively promoted seed germination at all times, but not up to the control level, suggesting that melatonin alleviates some of the damage caused by salt stress, not completely eliminates it. Meanwhile, the effect of high concentrations of melatonin (i.e., $MT_{50}$+NaCl and $MT_{100}$+NaCl treatments) on seed germination was not obvious, demonstrating that 20 μM melatonin was the optimum concentration for promoting cotton seed germination under salt stress (Fig 1), which indicates that the effect of exogenous melatonin is closely related to its concentration. A higher concentration of melatonin is not conductive to the mitigation of salt stress, and it is necessary to select an appropriate concentration of melatonin based on the specific situation. Similar results have been reported in rice and rapeseed [38, 39].

It has been suggested that melatonin improves the redox state of cells, thereby decreasing levels of ROS and reactive nitrogen species and stabilizing biological membranes in plant cells. Previous studies have shown that cold stress induced a considerable accumulation of hydrogen peroxide in cucumber and watermelon seeds, and exogenous melatonin can effectively inhibit the accumulation of hydrogen peroxide [40, 41]. Exogenous melatonin reduced MDA content in *Avena nuda* under salt stress and increased tolerance to salt stress [42], and an appropriate

concentration of melatonin was able reduce electrolyte exudation and effectively protect cell membranes in bermudagrass [43]. Additionally, melatonin-treated seedlings exhibited reduced oxidative damage through the inhibited overproduction of these ROS and MDA as well as EL. This may be owing to the role of melatonin in plants under abiotic stress, as it acts as an antioxidant that upregulates the expression of antioxidant enzymes, thereby reducing ROS levels [44]. In the present study, $H_2O_2$ was quite high in the control at 2 d, dropped drastically by 4 d, and rose again by 6 d. The amount of $H_2O_2$ in the salinity treatment at day 6 was less than that of the Control treatment at day 2, but the germination rate was significantly less than the control at all time points. The $MT_{50}$+NaCl and $MT_{100}$+NaCl treatments had substantially decreased $H_2O_2$ levels at 4 d and 6 d, but the germination rate was unimproved. Consequently, the amount of hydrogen peroxide may not be a particularly important variable, or the seed germination process may be quite tolerant to it (Fig 3). MDA and EL are important indicators of cell membrane stability. During the seed germination assay, salt stress led to membrane lipid oxidation and increased MDA and EL contents. However, the content of MDA and EL in cotton seeds decreased in response to melatonin treatment. The MDA and EL content of the 20 μM melatonin-treated seeds were significantly lower than those of NaCl-treated seeds, confirming that the effect of melatonin was dose dependent and that a suitable concentration of melatonin could reduce peroxidation damage to membrane lipids (Fig 4, Table 1). Based on these observations, an optimal melatonin concentration appears to mitigate the accumulation of $H_2O_2$, defending against oxidative stress, and too much melatonin disrupts ROS accumulation in germinating seeds, consistent with previous research [45].

The accumulation of osmotic regulators plays an important role in maintaining intracellular stability and protecting cells from salt stress and toxicity. Melatonin can promote the accumulation of proline in tomato seedlings under salt stress and also accelerates cucumber seed germination [44, 46]. In the present study, proline content dropped in the control over time, but increased under saline treatment by 2 d and then dropped to a similar level as the control by days 4 and 6, which indicates salt stress led to the accumulation of proline in the early stages of cotton seed germination. Proline content was increased by melatonin pretreatment, perhaps though melatonin regulating the related metabolism of osmotic substances and enhancing salt tolerance. Notably, $MT_{20}$+NaCl increases proline levels at all times, contributing to more normal osmotic balance in seeds. Notably, increased melatonin concentration does not lead to more proline (Fig 5A), indicating that excessive supplemental melatonin inhibits the accumulation of proline in germination seeds.

In the present experiment, the soluble sugar content overall decreased first and then increased in saline treatments, indicating that salt stress led to the accumulation of soluble sugar in cotton seeds. The sugar content was increased by intermediate concentrations of melatonin pretreatment ($MT_{20}$+NaCl and $MT_{50}$+NaCl) or decreased by low and high concentrations of melatonin pretreatment ($MT_{10}$+NaCl and $MT_{100}$+NaCl, respectively), which indicates that melatonin regulated the observed change in osmotic substances to improve salt tolerance (Fig 5B). The soluble protein content of cotton seeds decreased as caused by the degradation of proteins under salt stress, and the soluble protein content increased obviously after melatonin treatment, likely as a consequence of melatonin inducing protein synthesis and inhibiting the degradation process, thus maintaining the physiological activity and stability of cells (Fig 5C). In this study, exogenous application of 20 μM melatonin is most efficient in increasing the concentration of proline, soluble sugar, and soluble protein and these substances need to be greater in the experimental treatments than in the control to improve osmotic balance in salt stress seeds (Fig 5). However, even in the control, these substances vary with time, indicating that the germination of cotton seeds is a complex process, which is similar to previous results [38].

The dynamic balance of ions in plant cells plays an important role in plant growth and development, which can protect enzyme activity, and maintains membrane potential and osmotic pressure, thus maintaining cell volumes. Previous studies have demonstrated the potential of exogenous melatonin application in mediating $K^+/Na^+$ homeostasis and relative uptake rates of $K^+$ and $Na^+$ under salt stress in sweet potato [47]. Under salt stress, $Na^+$ and $Cl^-$ ions rush into the cells of seeds, causing the accumulation of $Na^+$, affecting the absorption of $K^+$ by plants, and causing ionic toxicity [48]. Maintaining a high $K^+/Na^+$ balance is essential to maintaining cell metabolism. Excessive salt ions disturb ion homeostasis and inhibit plant growth and development [49, 50]. Castanares et al. [36] concluded that salt stress leads to $K^+/Na^+$ imbalance, destroys cell membrane integrity, and reduces potassium retention, while exogenous melatonin effectively relieved this effect, which might be related to the activity of related enzymes in cells. Similarly, melatonin could significantly reduce $Na^+$ accumulation and increase $K^+$ content in maize seeds under salt stress [34]. Yu et al. [51] proposed a novel mechanism for melatonin-mediated salt tolerance, i.e., melatonin supplementation decreased the oxidative damage induced by salinity, perhaps by directly scavenging $H_2O_2$ or enhancing the activities of antioxidative enzymes. In addition, melatonin might control the expression of ion-channel genes (*MdNHX1* and *MdAKT1*) under salinity and maintain ion homeostasis and thus improve salinity resistance in plants exposed to exogenous melatonin [52]. As the germination time continued in the present experiment, the $Na^+$ content of cotton seeds increased continually, and the $K^+$ content of cotton seeds decreased and then increased in the Con and NaCl treatments. The $Na^+$ content of cotton seeds decreased obviously while $K^+$ content of cotton seeds increased obviously under salt stress. $Na^+$ content was significantly lower and $K^+$ content and $K^+/Na^+$ balance were significantly higher after melatonin treatments compared to Con seeds across different stages (Fig 6A–6C, Table 2). This demonstrated that the application of melatonin alleviated the accumulation of $Na^+$ by increasing $K^+$ absorption to maintain $K^+/Na^+$ balance and enhance salt tolerance. The present study also confirms the beneficial effect of 20 μM melatonin treatment on maintaining ion balance under salt stress.

## Conclusions

Exogenous melatonin was able to effectively alleviate salt stress damage and promote cotton seed germination by improving the physiological activity of cotton seeds by maintaining the $K^+/Na^+$ balance in vivo and promoting the content of melatonin and osmotic regulation substances. The 20 μM melatonin treatment was particularly effective in reducing the physiological damage caused by salt stress to cotton seeds and internally stabilizing cells, which was the most effective treatment in promoting seed germination under salt stress.

## Supporting information

**S1 File.**
(PDF)

**S1 Data.**
(XLSX)

## Acknowledgments

The authors thank the anonymous reviewers for their valuable comments and suggestions. We also thank our laboratory members for their help and efforts. Li Chen and Liantao Liu contributed equally to this research.

## Author Contributions

**Writing – original draft:** Li Chen, Liantao Liu.

**Writing – review & editing:** Bin Lu, Tongtong Ma, Dan Jiang, Jin Li, Ke Zhang, Hongchun Sun, Yongjiang Zhang, Zhiying Bai, Cundong Li.

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
