## [Decision Letter · Decision Letter 0]

1 Aug 2019

PONE-D-19-18506

Exogenous Melatonin Promotes Seed Germination and Osmotic Regulation under Salt Stress in Cotton (Gossypium hirsutum L.)

PLOS ONE

Dear Bai,

Thank you for submitting your manuscript to PLOS ONE. After careful consideration, we feel that it has merit but does not fully meet PLOS ONE’s publication criteria as it currently stands. Therefore, we invite you to submit a revised version of the manuscript that addresses the points raised during the review process.

We would appreciate receiving your revised manuscript by Sep 15 2019 11:59PM. To enhance the reproducibility of your results, we recommend that if applicable you deposit your laboratory protocols in protocols.io, where a protocol can be assigned its own identifier (DOI) such that it can be cited independently in the future. For instructions see: http://journals.plos.org/plosone/s/submission-guidelines#loc-laboratory-protocols

We look forward to receiving your revised manuscript.

Kind regards,

Sergey Shabala

Academic Editor

PLOS ONE

Journal Requirements:

1. Please include your tables as part of your main manuscript and remove the individual files. Please note that supplementary tables (should remain/ be uploaded) as separate "supporting information" files

Reviewers' comments:

Reviewer's Responses to Questions

**Comments to the Author**

1. Is the manuscript technically sound, and do the data support the conclusions?

Reviewer #1: Partly

Reviewer #2: Partly

2. Has the statistical analysis been performed appropriately and rigorously? 

Reviewer #1: No

Reviewer #2: No

3. Have the authors made all data underlying the findings in their manuscript fully available?

Reviewer #1: Yes

Reviewer #2: Yes

4. Is the manuscript presented in an intelligible fashion and written in standard English?

Reviewer #1: Yes

Reviewer #2: Yes

5. Review Comments to the Author

Reviewer #1: The manuscript entitled ‘Exogenous Melatonin Promotes Seed Germination and Osmotic Regulation under Salt Stress in Cotton (Gossypium hirsutum L.)” investigated the effect of melatonin in protecting cotton seeds under salinity and tried to explain its effect from the aspect of membrane integrity, ROS status, ion homeostasis. Although the study is well conceived and designed, its quality still needs to be improved. Please find the below comments for more details.

Major comments:

1. According to the significance letters on Figure 1, no significant difference of germination rate was found between salt stressed cotton plants with and without the melatonin treatment. Although it suggests a difference, an overlap of significance letters (for example, b and bc, or a and ab) between different groups indicates no significant difference at 0.05 level. Since only three replicates are conducted in this study, more replicates could help to improve the data accuracy. Or authors may redo the statistical analysis.

2. More in depth discussion will improve the quality of the current manuscript. For example, how melatonin reduced ROS content and maintained better K+/Na+ ratio in cotton seeds under salinity? Is it a direct response or indirect regulation? Does melatonin direct modulate the activity of ion channels and transporters? Or through reduce ROS content, melatonin indirectly affects ion transport.

3. Three replicates are the minimum requirement of biological studies. More replicates could improve the accuracy of data.

Minor comments:

Ln35. “the first evidence” is a bit over claim. For me, “evidence” is more related to something are clearly understood. Consider to revise it.

Ln 80-81. Cite (Mattuis 2014 J Exp Bot; Wu 2018, Crop J).

Ln330-331. Consider to rewrite it.

Ln 350. Consider to revise it. I would say “Na+ toxicity is one of the main component of salt stress in plants.”

Ln396. I would use “impaired” rather than “restricted”.

Ln396-397. Remove “Furthermore”.

Ln410. “affected” was repeated.

Ln427-428. “internal and external balance of cells”. It is not clear. Consider to revise or remove it.

Ln429. Over claim. No direct evidence showed in the current manuscript to support the claim that melatonin can directly scavenge ROS.

Ln459. “high K+ and low Na+” is not accurate. Under salt stress, Na+ accumulation and K+ loss happened. Maybe “to maintain K+/Na+ ratio” will be more accurate.

Figures. For better reading experience, according to the real setting of treatments described in the current manuscript, I would suggest to change “CK1” to “Con” or “Ctrl”, “CK2” to “NaCl”, “MT10” to “MT10 + NaCl”, “MT20” to “MT20 + NaCl”, “MT50” to “MT50 + NaCl”, and “MT100” to “MT100 + NaCl”.

Figure 2. The orders of significance labels are not appropriately presented. For example, in day2, “CK2” is the highest and is labelled with significance letter a, then the significance letter d should be assigned to the lowest one which is the “MT20” group. Also, please double check the assigned letters. Given the small error bar, “MT20” and “MT50” seem to have a significant difference.

Figure 5a. Why Na+ content in “CK2” group (150 mM NaCl) is decreased overtime? Also, in day6, “MT50” showed no protective role in preventing Na+ accumulation in cotton seeds under salinity.

Figure 5c. Please recheck the statistical analysis. Given the big error bars, most of the treatment showed no significant difference of K+ content. Another way is to increase sample replicates since only three replicates are conducted in the current manuscript.

Reviewer #2: The paper describes dose-dependent effect of exogenous melatonin on cotton seed germination under saline stress. The findings are interesting, but several aspects of the study need attention.

Introduction: Lines 57 – 59: there are relevant melatonin data and references, which should be included. Melatonin was introduced through cyanobacteria that became chloroplasts and mitochondria and is found in most plants and animals (Tan D-X et al. 2013, Mitochondria and chloroplasts as the original sites of melatonin synthesis: a hypothesis related to melatonin's primary function and evolution in eukaryotes. J Pineal Res 54: 127 – 138). So, there is likely to be endogenous melatonin in the cotton seed. Melatonin is extremely efficient antioxidant (Tan DX, Manchester LC, Terron MP, Flores LJ, Reiter RJ. One molecule, many derivatives: a never-ending interaction of melatonin with reactive oxygen and nitrogen species? J Pineal Res 2007; 42: 28 – 42; reference 90).

Line 64: Melatonin is closely related to IAA (indole -3-acetic acid) by structure and so are their metabolic pathways (see, for instance, Chapter 2, Auxin Biosynthesis and Catabolism, Yangbin Gao and Yunde Zhao, in Auxin and its Role in Plant Development, 2014, Eds. Zazimalova E, Petrasek J, Benkova E, Springer or Arnao MB and Hernandez-Ruiz J, 2018, Melatonin and its relationship to plant hormones, Annals of Botany 121: 195–207).

Methods: How was the salt stress level of 150 mM chosen? While the difference between salt stressed and control germination rates show clear significance, the effects of different melatonin concentration are very similar (especially comparing 10 and 20 μM data) in Fig. 1. Could greater saline stress give more distinct results?

I am also puzzled by the assignment of the significant difference (or lack there of) in some of the figures: for instance, in Fig. 2, at 6 d, MT50 and MT100 are surely not significantly different (but assigned letters “b” and “d”), while MT10 and MT20 are significantly different (but both assigned same letter “e”). There seem to be significance assignment problems in most of the figures (I am assuming that decreasing averages are assigned consecutive letters of the alphabet, if they are significantly different, depending on their standard deviations).

The units in some of the figures are not clear. Fig. 2: what does “mmol/g prot” mean? Fig. 3: microSiemens/cm? Fig. 4B: why not use microg? Fig. 4C: why is “fresh weight” added here – what about the Fig. 4A and B? Fig. 5A, B and C: why different units in A? Same type of units should be used for measurement of various substances, so the amounts can be compared (for instance amounts of proline and soluble sugars or Na, Cl and K).

It would be informative to measure endogenous melatonin content of the germinating seeds in control and under saline stress.

Results: While there are problems with the significance of the differences, the consistent trend of MT20 data does suggest, that this melatonin concentration is the most supportive one for germination in the face of saline stress.

Discussion: The authors should comment on: (1) The changes of the measured substances in the control and salt stressed seeds with time, which makes evaluating the role of melatonin more difficult, (2) Why is MT20 the most beneficial medium – the endogenous melatonin concentration might be helpful there.

Reference 20 and 32 seem to be the same paper.

6. PLOS authors have the option to publish the peer review history of their article (what does this mean?). If published, this will include your full peer review and any attached files.

Reviewer #1: No

Reviewer #2: Yes: Mary Jane Beilby

---

## [Author Response · Author response to Decision Letter 0]

27 Aug 2019

Responds to the Reviewer’s comments:

Reviewer #1

Major comments

1. According to the significance letters on Figure 1

---

## [Decision Letter · Decision Letter 1]

23 Sep 2019

PONE-D-19-18506R1

Exogenous Melatonin Promotes Seed Germination and Osmotic Regulation under Salt Stress in Cotton (Gossypium hirsutum L.)

PLOS ONE

Dear Bai,

The revised version of your MS has been reviewed again by the same set of reviewers. As you can see from their comments, several substantial concerns remain to be dealt with. I am willing to give you one more chance to address these. Please ensure that your review comprehensively deals with all reviewers' concerns as the number of iterations between reviewers and authors is limited. 

We would appreciate receiving your revised manuscript by Nov 07 2019 11:59PM. To enhance the reproducibility of your results, we recommend that if applicable you deposit your laboratory protocols in protocols.io, where a protocol can be assigned its own identifier (DOI) such that it can be cited independently in the future. For instructions see: http://journals.plos.org/plosone/s/submission-guidelines#loc-laboratory-protocols

We look forward to receiving your revised manuscript.

Kind regards,

Sergey Shabala

Academic Editor

PLOS ONE

Reviewers' comments:

Reviewer's Responses to Questions

**Comments to the Author**

1. If the authors have adequately addressed your comments raised in a previous round of review and you feel that this manuscript is now acceptable for publication, you may indicate that here to bypass the “Comments to the Author” section, enter your conflict of interest statement in the “Confidential to Editor” section, and submit your "Accept" recommendation.

Reviewer #1: (No Response)

Reviewer #2: (No Response)

2. Is the manuscript technically sound, and do the data support the conclusions?

Reviewer #1: Yes

Reviewer #2: No

3. Has the statistical analysis been performed appropriately and rigorously? 

Reviewer #1: (No Response)

Reviewer #2: No

4. Have the authors made all data underlying the findings in their manuscript fully available?

Reviewer #1: Yes

Reviewer #2: Yes

5. Is the manuscript presented in an intelligible fashion and written in standard English?

Reviewer #1: (No Response)

Reviewer #2: Yes

6. Review Comments to the Author

Reviewer #1: Authors have addressed most of my comments.

Few more comments

1. Please double check the English of the newly added text.

2. Add a vertical dash line between the dataset of different germination time points to indicate that the statistical analysis was separately done in each germination time point group.

3. Maathuis, 2014, Sodium in plants: perception, signalling, and regulation of sodium fluxes. Journal of Experimental Botany.

Reviewer #2: The manuscript is improved, but still marginal. The assignment of significance in many of the figures is still a problem: for instance in Fig. 1 the control at 2 days is surely significantly different from the control at 6 days, yet they are both assigned letter “a”! The “c” and “d” are not significantly different! You do want to compare amounts at different germination times.

The measurement of the melatonin content in the control and in the seeds under different treatments would make it easier to formulate the mechanism of action.

The germination rate increased over 6 days in the control. However, the various measured substances changed over this time. These trends need to be compared and discussed for different treatments in detail. For instance: hydrogen peroxide is quite high in the control at day 2, drops drastically in day 4 and rises again in day 6. In the salinity treatment the amount at day 6 is less than the control at day 2, but the germination rate is significantly less than the control at any time. The MT+50, MT+100 treatments decreased the hydrogen peroxide substantially at day 4 and 6, but have not improved the germination rate. Consequently, the amount of hydrogen peroxide may not be particularly important variable, or the seed germination process may be quite tolerant to it. The proline content drops in the control over time, but increases in saline treatment on day 2, but then drops to similar level as in control on days 4 and 6. MT+20 increases proline levels at all times, contributing to better osmotic state of the seed. It is interesting that increased melatonin concentration does not lead to more proline.

Discussion: All measured substances should be considered and compared to the time variation in the control and the saline stressed plants. So, the results need to be discussed in much greater detail with references to relevant tables and figures (e.g. lines 426 – 433). The general problems with salinity (lines 418 – 421) and the properties of melatonin (lines 435 – 439) were already described in the Introduction. The authors do not speculate why there is an optimal melatonin concentration. Reactive oxygen species (ROS) are also involved in signalling sequences. It is possible that too much melatonin disrupts these sequences in the germinating seed (however, this is not the case with hydrogen peroxide, which is higher at MT+50 and MT+100).

7. PLOS authors have the option to publish the peer review history of their article (what does this mean?). If published, this will include your full peer review and any attached files.

Reviewer #1: No

Reviewer #2: No

---

## [Author Response · Author response to Decision Letter 1]

25 Oct 2019

Dear Sergey Shabala

On behalf of my co-authors, we thank you very much for giving us an opportunity to revise our manuscript again, we appreciate editor and Reviewers very much for their positive and constructive comments and suggestions on our manuscript entitled “Exogenous melatonin promotes seed germination and osmotic regulation under salt stress in cotton (Gossypium hirsutum L.) (submitted as PONE-D-19-18506)”. Those comments are all valuable and very helpful for revising and improving our paper, as well as the important guiding significance to us researches. We have studied comments carefully and have made correction which we hope meet with approval. Revised portion are marked in red in the paper.

 The main corrections in the paper and the responds to the Reviewer’s comments are as flowing:

Responds to the Reviewer’s comments:

Reviewer #1

Few more comments

1. Please double check the English of the newly added text.

Response: We have made correction according to the Reviewer’s comments. Our manuscript has been carefully edited by a native English-speaking editor of MogoEdit, and the grammar, spelling, and punctuation have been verified and corrected where needed.

2. Add a vertical dash line between the dataset of different germination time points to indicate that the statistical analysis was separately done in each germination time point group 

Response: We have made correction according to the Reviewer’s comments. We have added a vertical dash line between the data at of different germination time points. Thank you for your suggestion.

3. Maathuis, 2014, Sodium in plants: perception, signaling, and regulation of sodium fluxes. Journal of experimental Botany.

 Response: We have made correction according to the Reviewer’s comments. We have added reference.

Reviewer #2

1. The manuscript is improved, but still marginal. The assignment of significance in many of the figures is still a problem: for instance in Fig. 1 the control at 2 days is surely significantly different from the control at 6 days, yet they are both assigned letter “a”! The “c” and “d” are not significantly different! You do want to compare amounts at different germination times.

Response: We have compared amounts at different treatments according to the Reviewer’s comments.

As for the assignment of significance in many of the figures, the meaning of the letter (a, b, c, d) is explained as follows:

In Fig. 1, firstly, we used SPSS software to conduct variance analysis on the data of different treatments (Control; NaCl; MT10+NaCl; MT20+NaCl; MT50+NaCl; MT100+NaCl) at 2 d, different letters (a, b, c, d) appeared in 2 d group. Then we used SPSS software to conduct variance analysis on the data of different treatments (Control; NaCl; MT10+NaCl; MT20+NaCl; MT50+NaCl; MT100+NaCl) on 4 d or 6 d respectively, different letters (a, b, c, d) also appeared in 4 d or 6 d group. So the control at 2 days is significantly different from the control at 6 days, yet they are both assigned letter “a”.

In addition, data from other figures and tables are analyzed in the same way.

2. The measurement of the melatonin content in the control and in the seeds under different treatments would make it easier to formulate the mechanism of action 

Response: We have made correction according to the Reviewer’s comments. We have supplemented the measurement of the melatonin content in the control and in the seeds under different treatments

3. The germination rate increased over 6 days in the control. However, the various measured substances changed over this time. These trends need to be compared and discussed for different treatments in detail. For instance: hydrogen peroxide is quite high in the control at day 2, drops drastically in day 4 and rises again in day 6. In the salinity treatment the amount at day 6 is less than the control at day 2, but the germination rate is significantly less than the control at any time. The MT+50, MT+100 treatments decreased the hydrogen peroxide substantically at day 4 and 6, but have not improved the germination rate. Consequently, the amount of hydrogen peroxide may not be particularly important variable, or the seed germination process may be quite tolerance to it. The proline content drops in the control over time, but increases in saline treatment on day 2, but then drops to similar level as in control on day 4 and 6. MT+20 increases proline levels at all time, contributing to better osmotic state of the seed. It is interesting that increased melatonin concentration does not lead to more proline.

Response: We have made correction according to the Reviewer’s comments. The detailed changes can be seen in lines 474-499 of the discussion.

4. Discussion: All measured substances should be considered and compared to the time variation in the control and the saline stressed plants. So, the results need to be discussed in much greater detail with referances to revelant tables and figures (e.g. lines 426-433). The general problems with salinity (lines 418-421) and the properties of melatonin (lines 435-439) were already described in the introduction. The authors do not speculate why there is an optimal melatonin concentration. Reactive oxygen species (ROS) are also involved in signalling sequences. It is possible that too much melatonin disrupts these sequences in the germination seed (however, this is not the case with hydrogen peroxide, which is higher at MT+50 and MT+100).

Response: We have made correction according to the Reviewer’s comments.The detailed changes can be seen in 502-525 of the discussion.

Special thanks to you for your good comments.

Sincerely yours,

Zhiying Bai

Email: zhiyingbai@126.com

---

## [Decision Letter · Decision Letter 2]

22 Nov 2019

PONE-D-19-18506R2

Exogenous Melatonin Promotes Seed Germination and Osmotic Regulation under Salt Stress in Cotton (Gossypium hirsutum L.)

PLOS ONE

Dear Bai,

Thank you for submitting your manuscript to PLOS ONE. After careful consideration, we feel that it has merit but does not fully meet PLOS ONE’s publication criteria as it currently stands. Therefore, we invite you to submit a revised version of the manuscript that addresses the points raised during the review process.

We would appreciate receiving your revised manuscript by Jan 06 2020 11:59PM. To enhance the reproducibility of your results, we recommend that if applicable you deposit your laboratory protocols in protocols.io, where a protocol can be assigned its own identifier (DOI) such that it can be cited independently in the future. For instructions see: http://journals.plos.org/plosone/s/submission-guidelines#loc-laboratory-protocols

We look forward to receiving your revised manuscript.

Kind regards,

Sergey Shabala

Academic Editor

PLOS ONE

Reviewers' comments:

Reviewer's Responses to Questions

**Comments to the Author**

1. If the authors have adequately addressed your comments raised in a previous round of review and you feel that this manuscript is now acceptable for publication, you may indicate that here to bypass the “Comments to the Author” section, enter your conflict of interest statement in the “Confidential to Editor” section, and submit your "Accept" recommendation.

Reviewer #1: All comments have been addressed

Reviewer #2: (No Response)

2. Is the manuscript technically sound, and do the data support the conclusions?

Reviewer #1: (No Response)

Reviewer #2: Partly

3. Has the statistical analysis been performed appropriately and rigorously? 

Reviewer #1: (No Response)

Reviewer #2: Yes

4. Have the authors made all data underlying the findings in their manuscript fully available?

Reviewer #1: (No Response)

Reviewer #2: Yes

5. Is the manuscript presented in an intelligible fashion and written in standard English?

Reviewer #1: (No Response)

Reviewer #2: Yes

6. Review Comments to the Author

Reviewer #1: Authors have addressed my comments properly. The English of this manuscript is polished. Overall, the quality of the manuscript is improved.

Reviewer #2: The measurements of melatonin content add to the information on the melatonin effect on the germination seeds – important increase of the dataset.

However, the results are still not discussed clearly and logically.

For instance: The germination rate increases with time. Similar pattern is observed in salt-stressed seeds, but with reduced rates. MT20 improves the rates at all times, but not up to control level. MT50 and 100 essentially leave the salt-stressed rate unchanged (Fig. 1). Melatonin levels decrease in the control (Fig.2). There is not sufficient data to decide if the seeds make less melatonin or equal (or greater) amounts, which get consumed by the antioxidant action. Lines 253-255: therefore it is not clear whether salinity stress “inhibits” melatonin synthesis. As there is less reactive oxygen species with more added melatonin, it is more likely that it is consumed by antioxidant action of the melatonin.

It is good to know that increased external melatonin translates to increased internal melatonin (Fig. 2). However, this increase in internal melatonin does not help to keep the germination rates up in case of MT50 and 100.

The M20 medium is most efficient in increasing the concentration of proline, soluble sugar and soluble protein and these substances need to be greater than in the control to improve osmotic balance in salt stressed seeds (Fig. 5). However, even in the control these substances vary with time.

In the Discussion, the authors need to distinguish between regulatory changes in the seed upon exposure to saline stress (such as increase in proline) or a change due to loss of control (such as increase in Na+).

So, the data still needs to be discussed in more consistent and logical manner.

7. PLOS authors have the option to publish the peer review history of their article (what does this mean?). If published, this will include your full peer review and any attached files.

Reviewer #1: No

Reviewer #2: No

---

## [Author Response · Author response to Decision Letter 2]

11 Dec 2019

Dear Sergey Shabala

On behalf of my co-authors, we thank you very much for your feedback and the opportunity to revise our manuscript again. We also thank the Reviewers very much for their positive and constructive comments and suggestions on our manuscript entitled “Exogenous melatonin promotes seed germination and osmotic regulation under salt stress in cotton (Gossypium hirsutum L.) (submitted as PONE-D-19-18506).” These comments were all valuable and very helpful in our efforts to improve our paper and also provided significant guidance in our research. We have considered the comments carefully and have made correction that we hope will meet with approval. The revised portions are marked in red in the paper.

 Below, the main corrections to the paper and responses to the Reviewer’s comments are provided.

Responses to the Reviewer’s comments:

Reviewer #2: The measurements of melatonin content add to the information on the melatonin effect on the germination seeds-important increased of the dataset.

However, the results are still not discussed clearly and logically.

For instance: The germination rate increases with time. Similar pattern is observed in salt-stressed seeds, but with reduced rates. MT20 improves the rates at all times, but not up to control level. MT50 and 100 essentially leave the salt-stressed rate unchanged (Fig. 1). Melatonin levels decrease in the control (Fig. 2). There is not sufficient data to decide if the seeds make less melatonin or equal (or greater) amounts, which get consumed by the antioxidant action. Lines 253-255: therefore it is not clear whether salinity stress “inhibits” melatonin synthesis. As there is less reactive oxygen species with more added melatonin, it is more likely that it is consumed by antioxidant action of the melatonin.

It is good to know that increased external melatonin translates to increased internal melatonin (Fig. 2). However, this increase in internal melatonin does not help to keep the germination rates up in case of MT50 and 100.

The M20 medium is most efficient in increasing the concentration of proline, soluble sugar and soluble protein and these substances need to be greater than in the control to improve osmotic balance in salt stressed seeds (Fig. 5). However, even in the control these substances vary with time.

In the Discussion, the authors need to distinguish between regulatory changes in the seed upon exposure to saline stress (such as increase in proline) or a change due to loss of control (such as increase in Na+).

So, the data still needs to be discussed in more consistent and logical manner.

Response: We have made corrections according to the Reviewer’s comments. The detailed changes can be seen in lines 440-451, 453-459, 475-477, 48-484, and 535-540 in the Discussion. Additionally, two new references have been added as support.

Special thanks to you for your good comments.

Sincerely yours,

Zhiying Bai

Email: zhiyingbai@126.com

---

## [Editor Report · Decision Letter 3]

13 Jan 2020

Exogenous Melatonin Promotes Seed Germination and Osmotic Regulation under Salt Stress in Cotton (Gossypium hirsutum L.)

PONE-D-19-18506R3

Dear Dr. Bai,

We are pleased to inform you that your manuscript has been judged scientifically suitable for publication and will be formally accepted for publication once it complies with all outstanding technical requirements.

With kind regards,

Sergey Shabala

Academic Editor

PLOS ONE
---

## [Editor Report · Acceptance letter]

15 Jan 2020

PONE-D-19-18506R3 

Exogenous Melatonin Promotes Seed Germination and Osmotic Regulation under Salt Stress in Cotton (*Gossypium hirsutum* L.) 

Dear Dr. Bai:

I am pleased to inform you that your manuscript has been deemed suitable for publication in PLOS ONE. Congratulations! Your manuscript is now with our production department. 

With kind regards,

on behalf of

Prof Sergey Shabala 

Academic Editor

PLOS ONE